# NuMA and Ninein: Dynein Cargo-Adaptors Without a Classical Cargo

**DOI:** 10.3390/cells14221797

**Published:** 2025-11-15

**Authors:** Keying Guo, Andreas Merdes

**Affiliations:** Centre de Biologie Intégrative, Université de Toulouse/CNRS, 118 Route de Narbonne, 31620 Toulouse, France; keying.guo@univ-tlse3.fr

**Keywords:** dynein, dynactin, NuMA, ninein, microtubule

## Abstract

Dynein is a minus-end-directed microtubule motor that transports a variety of cargoes. Cargo specificity is mediated by a class of adaptor proteins that bind to the interface between dynein and dynactin, along the length of the Arp1 filament of dynactin, and that co-activate the motor. NuMA, ninein, and ninein-like protein (Nlp) are cargo-adaptors that are involved in microtubule organization, rather than carrying portable cargoes. At the same time, ninein and Nlp are believed to be anchorage factors for gamma-tubulin ring complexes to the centrosome. Here, we discuss recent findings on the interaction of NuMA and ninein with the dynein/dynactin complex, and how these findings challenge earlier concepts on ninein-dependent microtubule organization via gamma-tubulin complexes. We do not intend to provide an encyclopedic review on NuMA and ninein, but rather develop a hypothesis about how conformational changes may regulate the activities and binding specificities of these two proteins.

## 1. Introduction

Cytoplasmic dynein-1 (subsequently referred to as “dynein”) is a motor that transports numerous cargoes in eukaryotic cells towards the minus-ends of microtubules. Dynein is a 1.4 MDa multiprotein complex of six different proteins, each present in two copies. The dynein heavy chain is the largest among these proteins, a force-generating subunit that belongs to the family of AAA+ ATPases, and it is linked via its tail domain to the other components: intermediate, light–intermediate, and light chains [1]. Dynein is bound to a 1.0 MDa activator complex, dynactin, that contains itself multiple copies of twelve different proteins [2]. The core of the dynactin complex is formed by a filament of the actin-related protein Arp1, which associates with the various other dynactin subunits, as well as with the tail of dynein [3]. In interphase, dynein can move such diverse cargoes as membrane vesicles, mitochondria, peroxisomes, melanosomes, aggresomes, and mRNAs in complex with ribonucleoprotein particles, as well as viruses [4]. During mitosis, dynein has been implicated in centrosome separation, spindle pole assembly, spindle orientation, and the silencing of the mitotic checkpoint [5].

## 2. NuMA as a Dynein Co-Activator and Cargo-Adaptor in Mitosis

Here, we want to focus on a role of dynein that does not involve the transport of cargo, as defined in a traditional sense: instead of moving cargo along a microtubule, dynein can also move the microtubule if the motor itself is immobilized at other cellular structures. A well-described example of such an arrangement can be found at the cellular cortex of mitotic cells, where the dynein/dynactin complex is attached to the plasma membrane by interacting with NuMA, cortical LGN, and Gαi (Figure 1) [6,7,8]. This interaction is conserved across many eukaryotic species, including *Drosophila* and *Caenorhabditis elegans* [9,10,11,12].

In this configuration, dynein can bind astral microtubules, whose plus-ends reach the cortex. The minus-end-directed motor activity of the dynein heavy chains creates pulling forces on astral microtubules that reorient the mitotic spindle. NuMA acts hereby as a co-activator of dynein/dynactin, and at the same time as a linker to LGN. The interaction with LGN is mediated by the carboxy-terminal domain of NuMA, and the contact with the dynein/dynactin complex is made by the amino-terminal 705 amino acids of NuMA [6,8]. In this configuration, a dimer of NuMA binds to dynein/dynactin by contacting the dynein light–intermediate chain via the NuMA HOOK domains (amino acids 1–158), and by aligning all along the dynactin core from the barbed end to the pointed end of the Arp1 filament, where the NuMA dimer remains in contact with its spindly motifs (amino acids 492–513; Figure 2) [13,14,15].

In vertebrate cells, an interaction between dynein/dynactin and NuMA is equally important for the formation of spindle poles, where the multiprotein complex contributes to the focusing of microtubules, with dynein providing minus-end-directed motility and NuMA binding to other microtubules via two microtubule binding motifs in its carboxy-terminal region [16,17,18]. Spindle microtubules bound to the NuMA carboxy-terminus can be considered cargoes, but also points of anchorage to create forces on microtubules that are contacted by the dynein heavy chains, pushing the latter towards the spindle equator. The force-producing activity of dynein is thereby dependent on the mitosis-specific phosphorylation of NuMA at multiple residues near its carboxy-terminus (Figure 3), in particular, at putative Aurora A phosphorylation sites S1969, S1991, and S2047 [15]. In the absence of phosphorylation, an auto-inhibition of NuMA seems to take place through an interaction between the carboxy-terminal and the amino-terminal regions [15]. Although not shown experimentally, it is conceivable that the NuMA carboxy-terminus folds back towards the amino-terminal region, thus suppressing the activation of the dynein/dynactin complex (Figure 3). The mitotic microtubule-organizing activity of the NuMA/dynein/dynactin complex is further enhanced by the clustering of the supercomplex, mediated by the NuMA tail domain (Figure 4) [19,20,21].

## 3. Ninein and NLP as a Dynein Co-Activators and Cargo-Adaptor During Interphase

While NuMA is a prototype of a dynein co-activator and microtubule organizer in mitosis, the proteins ninein and ninein-like protein (Nlp) may be considered functionally related proteins that organize microtubules, but mainly during interphase [22]. Microtubule-organizing proteins related to ninein and Nlp have been identified in vertebrates and in several non-vertebrate species [23,24,25,26]. Similarly to NuMA, ninein and Nlp possess a dynein/dynactin-binding domain in the amino-terminal region. Unlike NuMA, ninein does not possess a HOOK domain, but establishes contact with dynein light–intermediate chains near the dynactin core via a pair of amino-terminal EF hands (Figure 4) [14].

An alignment along the dynactin Arp1 filament and contact at the barbed end via a spindly motif (amino acids 543–564) are features that ninein shares with NuMA and various other dynein co-activators [14,27]. However, NuMA and ninein differ significantly in their carboxy-terminal regions (Figure 4): whereas NuMA contains two microtubule-binding motifs; the ninein tail is lacking such motifs, but has a binding site at amino acids 1933–2096 that mediates the interaction with the centrosomal protein CEP250 (also known as C-Nap1) [28]. This binding site is used to anchor ninein to the proximal ends of both centrioles in the interphase centrosome (Figure 5A) [29]. At this location, ninein is thought to contribute to the cohesion between the two centrioles [30]. Although the mechanisms of ninein-dependent cohesion remain unknown, it is imaginable that the minus-end-directed motility of the ninein/dynein/dynactin complex on microtubules that originate from the opposing centriole would keep the two centrioles at a close distance. The centrioles could thus be considered cargoes (Figure 5A). Besides facilitating centrosome cohesion, centrosome-bound ninein/dynein/dynactin may enable the clustering of multiple centrosomes in syncytial cells, such as osteoclasts [31,32].

The bulk of ninein concentrates at specialized structures of the mother centriole: the subdistal appendages [33]. The recruitment of ninein to the subdistal appendages may be mediated by proteins ODF2, trichoplein, and CCDC120, but the exact mechanisms remain unclear [34,35]. Ninein itself is responsible for the recruitment of another protein, CEP170, to the subdistal appendages via an interaction domain encoded by exon 18 (amino acids 807–1518) [28]. The subdistal appendages have been proposed to anchor a subset of centrosomal microtubules and contribute to their radial organization [36,37].

## 4. Mechanisms of Microtubule Organization by Ninein

It is not fully understood how centrosome-bound ninein organizes radial microtubules. Two principal mechanisms come to one’s mind: (I) ninein could bind to microtubule minus-ends via the gamma-tubulin ring complex (gamma-TuRC), or (II) ninein could bind to microtubules via dynein as part of the ninein/dynein/dynactin complex.

(I) The first possibility is supported by the finding that the amino-terminal end of ninein can bind to gamma-tubulin and GCP3 via amino acids 1–246 [29]. Similar binding properties have been reported for the amino-terminal half of Nlp (mapped to amino acids 1–702) [38]. The problem with such a model is that the amino-terminus of ninein appears to be tightly shielded as long as ninein is part of the dynein/dynactin complex. For steric reasons, it is difficult to assume that a structure as voluminous as gamma-TuRC (25 nm diameter) can access ninein in such a configuration. Another paradox is the finding by [39] that gamma-tubulin can bind to ninein residues 1179–1931, a region far away from the ninein amino-terminus. However, it would be imaginable that ninein binds to the gamma-TuRC after a conformational change via a composite binding surface of amino acids 1–246 and 1179–1931, and without being associated with dynein/dynactin (Figure 5B). Consistently, it has been reported that ninein amino acids 1181–2096 can interact with a region of amino acids 461–1193 [40]. For this, a hinge region in the coiled-coil-forming part of ninein would have to allow a refolding of the ninein molecule. Yet, the refolding of ninein may mask the carboxy-terminal CEP250-binding domain, in which case the anchorage to the centrosome would have to be mediated by different contact sites. Regulators that could influence the conformation of ninein have been proposed to include kinases GSK3beta [28], as well as Aurora A and protein kinase A [40].

(II) The second model for ninein-dependent microtubule organization, via contacts by the dynein motor, would require the first ~600 amino acids of ninein to be aligned with dynein/dynactin, free from any other interaction. In particular, ninein amino acids 1181–2096 would need to be disconnected from amino acids 461–1193. It may be speculated whether the centrosomal binding partners that interact with the carboxy-terminal end of ninein contribute to the disconnection between the amino- and carboxy-terminal halves of ninein and thereby activate the formation of the ninein/dynein/dynactin complex. This would create a localized activity on free microtubules, pushing them outwards from the centrosome, with the plus-ends pointing towards the cell periphery (Figure 5C), and it would lead to the self-centering of the centrosome when microtubule plus-ends hit resistance at the cellular cortex. Similar mechanisms, involving immobilized protein complexes of ninein/dynein/dynactin, appear to be employed for microtubule organization from non-centrosomal sites (Figure 6). These include perinuclear microtubule organization in muscle cells or in fat body cells in *Drosophila*, where ninein is bound to the outer nuclear envelope [32,41,42], cortical microtubule organization in epidermal keratinocytes, where ninein is immobilized at desmosomes by desmoplakin [43,44,45], and cortical microtubules in *Drosophila* oocytes [46]. It would be interesting to determine whether the binding of ninein to these non-centrosomal organizing centers involves alterative splicing, resulting in ninein isoforms with variable carboxy-terminal domains [47].

## 5. Conclusions

NuMA and ninein are both co-activators and cargo-adaptors of the dynein/dynactin complex, but with untypical cargoes. Instead of mediating the transport of membrane-bounded organelles, both proteins can play roles in the translocation of microtubules, powered by the dynein motor activity, while being immobilized at other cellular structures. Numerous aspects still need experimental investigation, such as the mechanism of activation of NuMA and ninein and the importance of conformational changes in both proteins. In particular, the mode of interaction of ninein with its binding partners needs to be studied, and the hypothesis of the displacement of the dynein/dynactin complex from binding to the gamma-TuRC needs experimental verification. Also, it would be useful to know whether ninein binds preferably to either gamma-TuRCs or dynein/dynactin at subdistal appendages or at the proximal ends of the centriolar cylinders, and whether a loss of the cap provided by the gamma-TuRC at microtubule minus-ends can induce the motor activity of ninein/dynein/dynactin.

## Figures and Tables

**Figure 1 cells-14-01797-f001:**
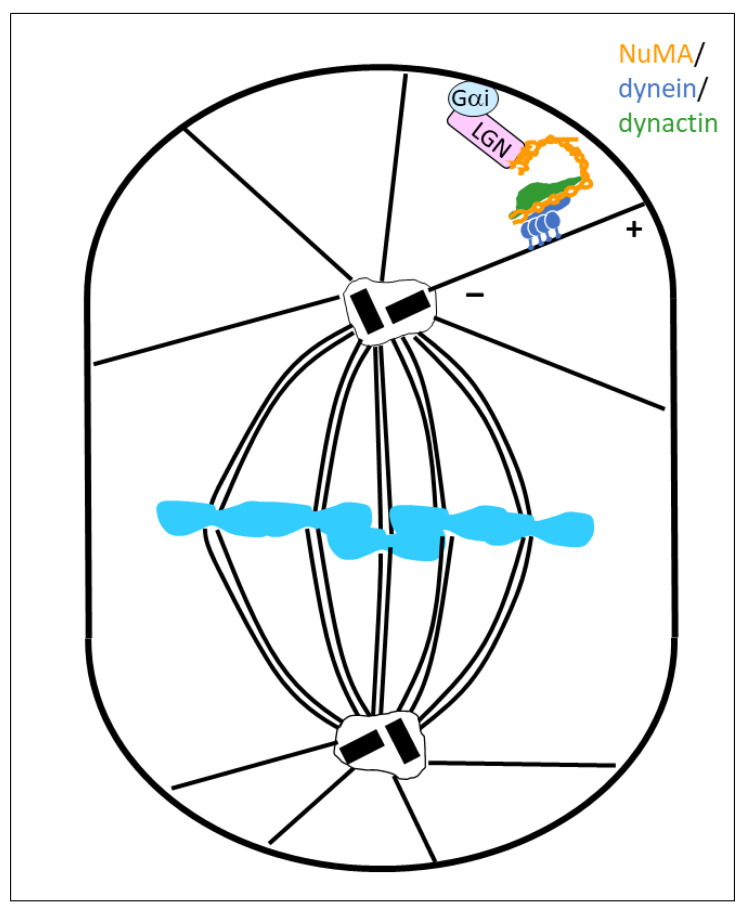
Spindle orientation, driven by a cortically anchored NuMA/dynein/dynactin complex. A plasma membrane-bound Gαi protein binds the LGN C-terminal end, whereas the LGN N-terminus binds to the NuMA tail domain. The NuMA-bound dynein/dynactin complex exerts pulling forces on astral microtubules that drive reorientation of the mitotic spindle.

**Figure 2 cells-14-01797-f002:**
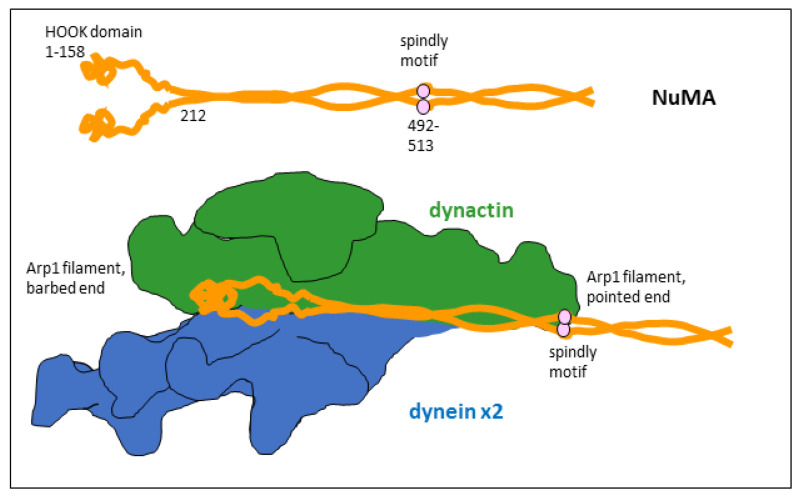
The NuMA/dynein/dynactin complex. Top: domain organization in a dimer of NuMA. Only the amino-terminal third of NuMA is shown. The spindly motif is a conserved stretch of amino acids in various dynein/dynactin co-activators. Bottom: simplified model of the supercomplex of NuMA, two copies of dynein, and dynactin (drawing inspired from [15]). The amino-terminal parts of a NuMA dimer align with the Arp1 filament of dynactin.

**Figure 3 cells-14-01797-f003:**
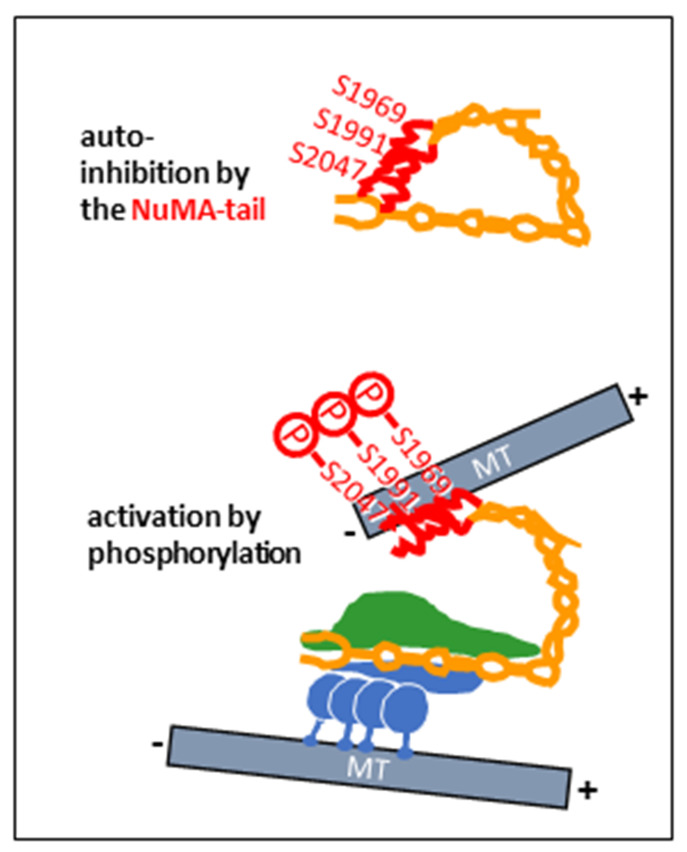
Hypothetic model of conformational changes in NuMA, leading to the activation of the dynein/dynactin complex. Top: NuMA is auto-inhibited by the NuMA tail (red), folded back towards the amino-terminal region of the protein. Bottom: mitotic phosphorylation of NuMA tail amino acids S1969, S1991, and S2047 activates the dynein/dynactin complex through conformational changes in NuMA. The carboxy-terminal tail of NuMA possesses microtubule-binding regions that have an affinity for microtubule minus-ends. Minus-ends can thus be pulled and tethered at the spindle pole by NuMA/dynein/dynactin.

**Figure 4 cells-14-01797-f004:**
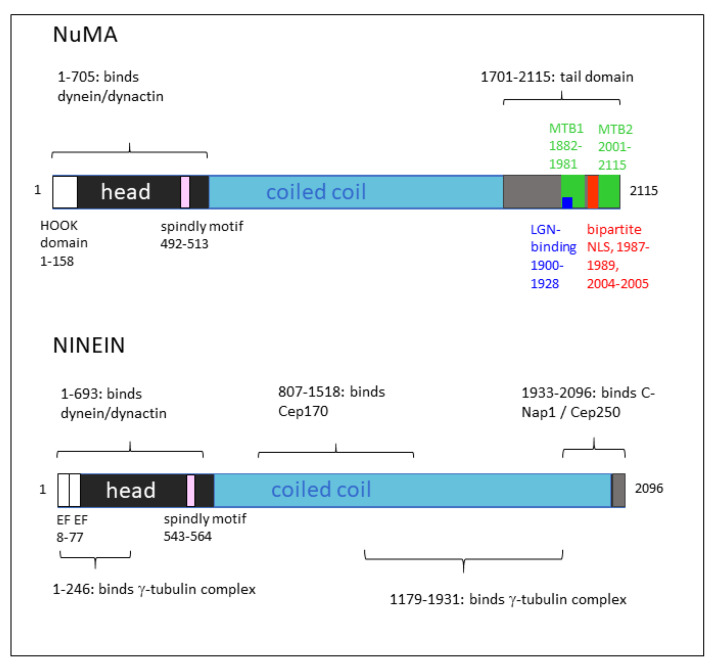
Domain organization of NuMA and ninein. The amino acid positions of various interaction domains, such as a HOOK domain, a spindly motif, two microtubule-binding domains (MTB1 and MTB2), an LGN-binding domain, and a nuclear localization sequence (NLS), are indicated for NuMA. In ninein, the amino-terminal position of a pair of EF hand motifs, the localization of the spindly motif, and regions that interact with centrosome proteins are indicated.

**Figure 5 cells-14-01797-f005:**
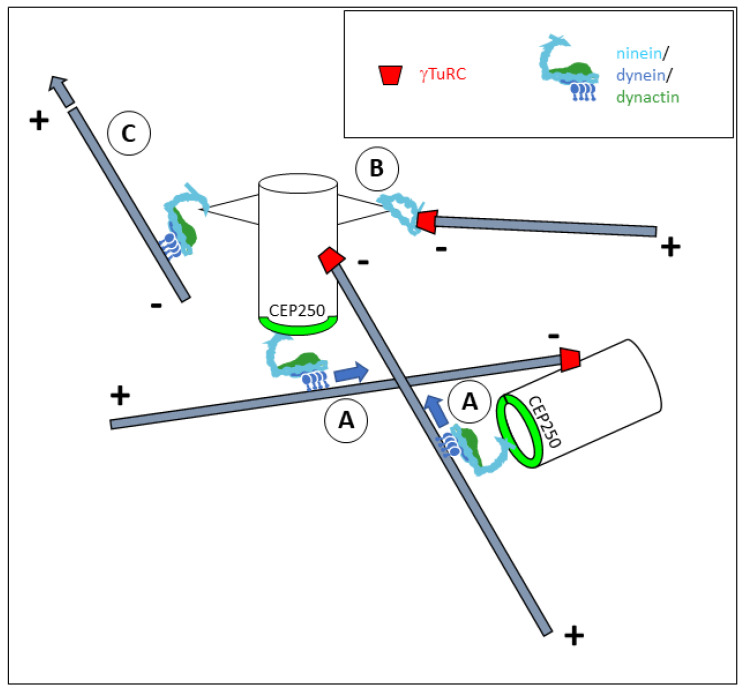
Scheme, indicating the localization and activity of ninein at the centrosome. Mother and daughter centrioles have the centrosome protein CEP250 localized to the proximal ends of their cylinders (light green). (A) The carboxy-terminal ends of ninein anchor ninein/dynein/dynactin to CEP250. The minus-end-directed motor activity of dynein can move the centriolar cylinders along microtubules emanating from each opposite centriole, thereby bringing mother and daughter centrioles close together (“centriole cohesion”). (B) At the subdistal appendages of the mother centriole, ninein may anchor gamma-tubulin complexes in the absence of dynein/dynactin, potentially via a composite binding region of amino acids 1–246 and 1179–1931 (see Figure 4). This may allow the direct anchorage of microtubule minus-ends. (C) Ninein can anchor microtubules via associated dynein/dynactin, and can produce forces on these microtubules, with the plus-ends projecting towards the cell periphery.

**Figure 6 cells-14-01797-f006:**
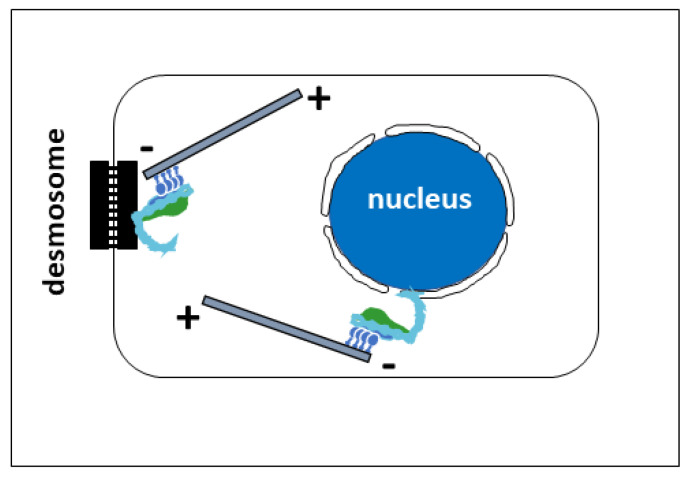
Microtubule organization by ninein at non-centrosomal sites. Left: in keratinocytes, desmosomes can anchor ninein/dynein/dynactin through an interaction between desmoplakin and ninein. The complex is involved in cortical microtubule organization. Right: in muscle cells and in *Drosophila* fat body cells, ninein is bound to the outer nuclear membrane and may be involved in microtubule organization from the nuclear surface.

## Data Availability

No new data were created or analyzed in this study.

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
