# Peer review of "NuMA and Ninein: Dynein Cargo-Adaptors Without a Classical Cargo"

_cells, 2025, doi:10.3390/cells14221797_

Round 1
Reviewer 1 Report
Comments and Suggestions for Authors
This Opinion piece is timely and useful: it synthesizes new structural and cell-biological insights on NuMA and ninein as dynein/dynactin co-activators that organize microtubules without ferrying canonical “cargo,” reframing how readers think about spindle orientation, pole focusing, centrosomal and non-centrosomal MTOCs, and conformational regulation. The narrative anchors recent advances (LIC–adaptor interfaces, spindly motifs, Arp1 alignment) to clear, didactic schematics and proposes testable models. Notably, the figures employ simplified schematics—rather than detailed protein models—which makes the pathway logic and domain relationships easy to follow and lowers the barrier for non-specialists. With a few corrections—the manuscript will serve the Cells audience well. I support publication pending minor revisions.
Obvious errors to fix (callouts, captions, text, references)
- Page 2 line 46 G(alpha)i
- Figure 2—typo in caption. “supercomlex” → supercomplex. (Also ensure consistent capitalization: “NuMA/dynein/dynactin complex.”)
- Figure 5—typo. “centromal microtubules” → centrosomal microtubules; also check “centriole cohesion” phrasing for consistency (no split words across lines).
- Terminology consistency. Use one of “microtubule minus-ends are pulled/tethered” instead of alternating “anti-poleward force” versus “ejection forces” unless you define both terms
- Please include a side-by-side, domain map for NuMA (NUMA1), Ninein (NIN), and if possible Ninein-like protein showing the major regions each uses to engage dynein/dynactin and microtubules—e.g., extended coiled-coil segments, dynein/dynactin-recruitment motifs (spindly-like/activating elements), centrosomal targeting regions, and any MT-binding or EF-hand–like segments—annotated with residue ranges and a common legend/color key.
6. NuMA phosphoregulation link to autoinhibition. You cite specific tail phosphosites (S1969, S1991, S2047) in Fig. 3; consider adding one in-text sentence tying these sites to the kinases responsible for this PTM.
Reviewer 2 Report
Comments and Suggestions for Authors
Guo and Merdes present a succinct and coherent Opinion on the roles of NuMA and Ninein in the regulation of Dynein activity and microtubule organization. The manuscript is well written and offers a contemporary summary of how these two regulators represent unique modes of Dynein interaction to control microtubule function. I have a few minor suggestions for how the manuscript might be improved. Otherwise, I support publication of the manuscript.
- The Figure quality and aesthetics could be improved. This is mostly stylistic, but the oversimplified cartoons also suffer from not conveying sufficient information in some cases as well (e.g. Figure 2, bottom). I would recommend attempting to improve their quality, both stylistically and in the detail they provide the reader.
- It might be beneficial to comment on which NuMA and Ninein functions are conserved in other model organisms (e.g. flies, worms) to provide readers with a perspective on the evolutionary importance of these functions.
- If relevant to their models, the authors might consider adding references and a short discussion of the NuMA clustering domain and its role in microtubule regulation (Okumura M et al 2018; Chinen T et al 2020).
- In Figure 2, it would be helpful to have a full-length NuMA domain diagram similar to that presented in Figure 4 for Ninein.
